# Multi-assay approach shows species-associated personality patterns in two socially distinct gerbil species

**Andrey V. Tchabovsky***, **Elena N. Surkova, Ludmila E. Savinetskaya**

Laboratory for Population Ecology, Severtsov Institute of Ecology and Evolution, Russian Academy of Sciences, Moscow, Russia

* tchabovsky@gmail.com

**Data Availability Statement:** All relevant data are within the manuscript and its Supporting information files.

## Abstract

We aimed to investigate whether two closely related but socially distinct species of gerbils differ in personality patterns. Using a suit of multivariate repeated assays (docility test, dark-light emergence test, startle test, novel object test, elevated platform test, and stranger test), we assessed contextual and temporal consistency of docility, boldness, exploration, anxiety, and sociability in the solitary midday gerbil, *Meriones meridianus*, and social Mongolian gerbil, *M. unguiculatus*. We revealed contextually consistent and highly repeatable sex-independent but species-specific personality traits. Species differed in temporal repeatability of different behaviours, and contextual consistency was more pronounced in solitary *M. meridianus* than in social *M. unguiculatus*. This finding contradicts the social niche specialization hypothesis, which suggests that personality traits should be more consistent in more social species. Instead, we hypothesize that social complexity should favour more flexible and less consistent behavioural traits. The habituation effect indicative of learning abilities was weak in both species yet stronger in social *M. unguiculatus*, supporting the relationship between the sociality level and cognitive skills. In both species, only a few different behavioural traits covaried, and the sets of correlated behaviours were species-specific such that the two species did not share any pair of correlated traits. Between-species differences in personality traits, habituation, and behavioural syndromes may be linked to differences in sociality. The lack of prominent behavioural syndromes is consistent with the idea that context-specific individual behavioural traits might be favoured to allow more flexible and adequate responses to changing environments than syndromes of correlated functionally different behaviours.

## Introduction

Animal personality is defined as the inter-individual variation in behavioural traits consistent over time and across different contexts [1–3]. In other words, an individual's behaviour must be repeatable and vary temporally and contextually less within than among individuals to be considered a personality trait [4]. Moreover, different seemingly unrelated behaviours can also

**Funding:** This research was supported by the Russian Science Foundation (22-14-00223 to AVT, https://rscf.ru/project/22-14-00223/) The funders had no role in study design, data collection and analysis, decision to publish, or preparation of the manuscript.

**Competing interests:** The authors have declared that no competing interests exist.

covary among and within individuals, creating behavioural syndromes, which suggests that they share proximate mechanisms and are governed by a common background trait [1, 5–7, but see 8, 9].

Personality traits were found to be ecologically and evolutionary relevant and can influence (or be related to) the individual life history [10], e.g., dispersal [11], space-use [12], gut microbiota [13], habitat selection [14], fitness and survival [15, 16], as well as group- and population-level processes such as social organization [17], migration rate, population fluctuations [18], spatial population dynamics [19], interspecies competition [20], species range expansion, colonization, and invasions [21–23]. Unsurprisingly, in the past two decades, animal personality research was integrated into most organismal biological fields providing new insights into animal ecology and evolution, applied sciences, and human behaviour [3].

Many personality traits have been described in vertebrates and invertebrates, but most studies have focused on broad categories of behaviour: boldness/shyness, exploration, activity, aggressiveness, and sociability proposed by Réale et al. [2, 3]. In the laboratory and the field, these behaviours are mainly investigated experimentally by measuring inter-individual variation in behavioural responses to the more or less standardized experimental situation. The most common assays include the open field test, dark-light emergence test, startle test, novel object test, elevated plus-maze test, novel conspecific test, and docility test [2, 3, 12, 15, 24–30]. All these tests are quick, convenient, flexible in setup, and easy to repeat, making them useful tools for researchers; nevertheless, there is a growing awareness of the need to validate standardized tests initially developed for domesticated laboratory rodents for non-model species of wild rodents [12, 29].

Repeated measurements are a prerequisite for validating the personality tests, although single assays conducted only once per individual are common [4, 8, 31, 32]. To assess the temporal and contextual consistency of individual responses (i.e. the key attributes of personality), the same individuals must be measured repeatedly within a context and across contexts, as this is the only way to estimate the contribution of among-individual differences to behavioural variation and establish whether or not individuals consistently differ [4, 33]. In particular, to estimate contextual consistency, personality tests should measure the same or similar behaviours (associated, for example, with boldness) in multiple assays in different contexts [31, 34]. Using a single assay assumes it reflects the same behavioural trait in other contexts; nevertheless, this assumption may be false [35].

Personality and behavioural syndromes may vary or not between sexes [36, 37]. For example, in chimpanzees, males differed from females in most measured social personality traits [38]. Conversely, in house mice, females did not differ in their exploratory behaviour from males [39]. In *Peromyscus* (deer mice), sexual differentiation was observed only in sociability but not in boldness, activity, or exploratory behaviour and only in non-monogamous species [27]. Therefore, the effect of sex on behavioural traits may vary across species and should be included in the models testing personalities [8].

Intra- and interspecies variation in individual behavioural traits is the key to understanding the ecological and evolutionary relevance of personality [12, 40]. Individual responses to the same test situation may vary between species or populations [34, 40, 41]. For example, wild house mice (*Mus musculus domesticus*) from two populations differed in activity/exploration but not in anxiety-like behaviour [12]. Ecologically relevant differences in the consistency and flexibility of behavioural traits were found among four species of Australian funnel-web spiders [40]. Between-species differences in behavioural syndromes related to the differences in mating systems were revealed in deer mice [27]. "The social niche specialization" hypothesis [42] predicts that the consistency of personality traits should vary with the species' sociality, with more social species exhibiting greater between-individual differences. Nevertheless, thus

far, only a few studies have compared personality consistency between closely related but socially different species [27, 43, 44]. Thus, species specificity should be considered when interpreting the results of personality tests [8], and the interspecies comparisons under standardized test procedures are in high demand as they can provide new insights into ecological and evolutionary drivers of variation in behavioural traits within and between species [12, 27, 40, 43, 45].

In this work, we aimed to investigate whether two closely related but socially distinct species of gerbils differ in personality patterns. Using standardized repeated tests with multiple behavioural measurements obtained in different contexts, we studied whether individual behaviours (docility, boldness, exploration, anxiety, and sociability) are correlated and consistent across contexts and time and if they are species-specific. We show that both gerbil species exhibit temporally and contextually consistent individual behavioural traits (i.e., personalities). Moreover, the patterns of behavioural consistency and behavioural syndromes differed between species following between-species differences in sociality. This is the first comparative analysis of personalities in gerbils—the model taxon for comparative studies in the various fields of behavioural science and ecology—and, to our knowledge, the third one that relates personality in mammals to the species' sociality.

## Materials and methods

### Study animals and housing

We used two non-model species of rodents—the midday (*Meriones meridianus* Pallas, 1771) and the Mongolian (*M. unguiculatus* Milne-Edwards, 1867) gerbils. They are closely related, similar in size (50–70 g of adult body weight) and ecology, but differ drastically in social and mating systems. Both species live in open landscapes of arid and semiarid zones and sympatric in some areas of their geographical ranges [46]. *M. meridianus* is a nocturnal-diurnal granivorous/folivorous psammophilous gerbil inhabiting deserts and semi-deserts of Central Asia, Northern China, and Southern Russia [47–50]. Midday gerbils live solitarily or in loose aggregations: males and females are not territorial, do not form stable pair bonds, interact rarely, and display little agonistic or amicable behaviour [51, 52]. *M. unguiculatus* is the most closely related species to *M. meridianus* [53], but socially different [52, 54]. This diurnal folivorous/granivorous gerbil inhabits semi-arid steppes and desert grasslands of Inner Mongolia of China, Mongolia, and Russia [55]. It is one of the most social species within the subfamily Gerbillinae [52, 54, 56, 57]: they live in complex family groups, an adult male and a female share a nest, form a stable pair bond, take care of young, and defend the group territory from intruders by exhibiting intra- and intersexual stranger-directed aggression. The pairwise comparison of closely related species is often the best tool for relating species differences in a given pattern to the social systems (e.g. [58]) or any other contrasting traits expected to affect the variable of interest because few confounding factors are likely to influence the conclusions [59].

Young midday gerbils (seven males and five females) were taken as weanlings from a natural population in Kalmykia, Southern Russia. Young Mongolian gerbils (four males and five females) were from the captive-bred colony of the Moscow Zoo, descending from wild-caught gerbils collected in Tuva (Russia) in 2008, and outbred to maintain genetic variation, which allows investigating species-typical behaviour in a standardized environment [27]. Post weaning, the young of both species were reared in standardized conditions in the laboratory.

The main reason for the small sample is the low availability of wild non-model species used in our study compared to highly available traditional laboratory animals—a common problem in behavioural studies where the average $N = 21$ for mammals [60]. Nevertheless, studying

non-model species is essential to infer the ecological or evolutionary relevance of personality traits [12].

Before the experiment, gerbils were housed individually for two months in standard plastic cages 50 x 30 x 20 cm with wire ceiling on pine shaving bedding with a wooden box as a shelter, some hay, small branches and pine cones for enrichment, food (grain, carrots, apples) available ad libitum and no water provided because gerbils do not drink in nature and obtain water from their food, under natural light-dark regime at room temperature. All animals used in the experiments were adults (post-natal day 90–150), all males had scrotal testis, and females showed no signs of receptivity.

## Experimental procedure

The entire procedure (the trial) included four consecutive stages (tests), which assessed the same as well as different behavioural measurements in different contexts (Table 1, Fig 1). The first test evaluated a single measurement—the activity of a gerbil in a handling bag. The second, third, and fourth tests were composite, included multiple measurements, and were carried out in suits of consecutive assays. All animals ($N = 21$) were tested in the same order of assays in the same experimental compartment, which increases the validity of personality tests by standardizing the experimental environment [8, 9]. Each animal passed the trial twice (the most common and recommended approach to measure behavioural repeatability–[29, 61]) at a 3-day interval.

**(A) Bag test.** A small cotton handling bag (35 x 25 cm) with a gerbil inside was placed in a plastic bowl, and the time (in seconds) the gerbil spent immobile during 1 min was measured. This test was used to assess docility (response to restraint) as a personality metric in rodents and other small mammals [15, 24, 26, 28].

**(B) Dark-light/startle/novel object test (DL/S/NO).** After the bag test, the animal was released into an opaque metal box (25 x 7 x 7 cm) with a wire-mesh bottom. The box was

**Table 1. The sequence of tests, recorded behaviours, associated traits, and measurements.**

| Test | Behaviours | Behavioural traits | (#) Measurements |
|---|---|---|---|
| **(A) Bag** | Immobility | Docility | (1) Time immobile (s) |
| **(B) Dark-light/Startle/Novel object (DL/S/NO)** | | | Latencies (s): |
| DL S | Emergence from the shelter | Boldness | (2) HO (3) BO |
| NO | Sniffing object | Exploration | (4) CO |
| **(C) Dark-light/Elevated platform (DL/EP)** | | | Latencies (s): |
| DL | Emergence from the shelter | Boldness | (5) HO (6) BO |
| EP | Climbing down | Anxiety | (7) CD |
| **(D) Dark-light/Stranger (DL/STR)** | | | Latencies (s): |
| DL | Emergence from the shelter | Boldness | (8) HO (9) BO |
| STR | Sniffing stranger | Sociability | (10) CS |

Three composite tests (B, C, D) included two same behavioural measurements (HO and BO) and one test-specific measurement—the latency to the final event (FE) (see Fig 1).

HO—head out, BO—body out. FEs: CO—contact object, CD—climb down, CS—contact stranger. Note that Boldness, Exploration, and Sociability were measured with the opposite to the traits metrics—the latencies to events (HO and BO, CO, and CS, respectively), so that the shorter the latencies, the bolder, more explorative, and more sociable the individuals are. Docility and Anxiety were measured with the direct metrics—the more time immobile and the longer the latency to CD, the more docile and anxious the individual, respectively.

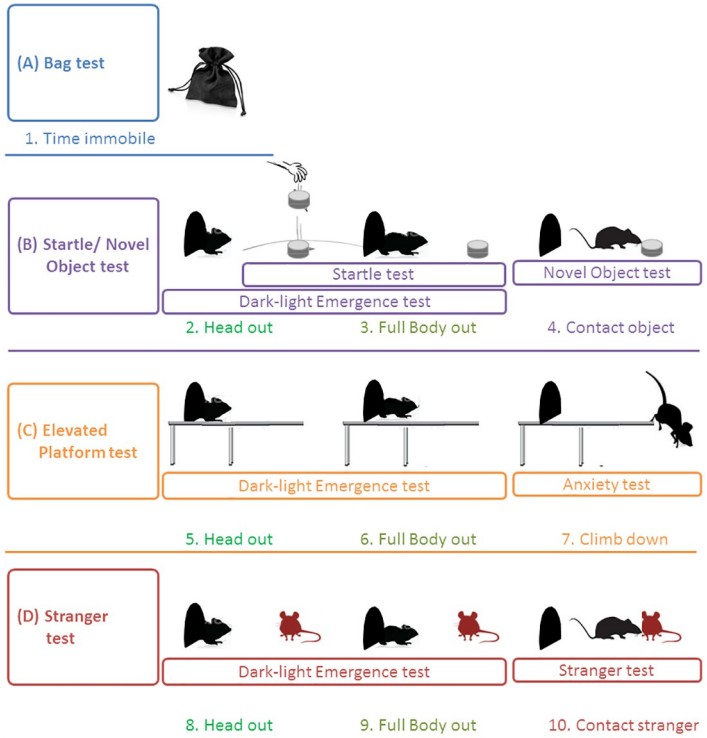

**Fig 1. The scheme of experimental setup.** Measurements in the same color show the same measurements in different tests (contexts). Measurements in different colors show different measurements within the test. Test-specific measurements (final events, FE) are indicated by corresponding test-specific color: blue for the Bag test, violet for the Startle/Novel object test, orange for the Elevated platform test, and red for the Stranger test (see Table 1).

placed at the short side of the experimental illuminated plastic arena (78 x 56 x 43 cm) with a floor covered with sand; the door of the box looked toward the center. After 1 min of acclimation, the door was opened distantly, and the Dark-light (DL) part of the test started. When the gerbil showed its head from the box ("head out"–HO), the Startle (S) part of the experiment began, and a metal cylinder (5 x 4 cm) filled with metal balls (total mass = 165 g) was dropped with noise at the opposite side of the arena, from a height of 45 cm. Gerbils either responded to the startle noisy stimulus by retreating to the box or did not, staying at the box entry. The latency from the start of the experiment until the gerbil emerged from the box with all four limbs ("full-body out"–BO) was measured. After that, without interruption, the Novel object (NO) part of the test began, and the time it took the gerbil to approach and sniff the dropped cylinder ("contact novel object"–CO) was measured. If an animal did not come out after being startled or did not approach the object during the test period of 5 min, the latencies were set to 300 s. This test was adapted from the tests for boldness and exploration used in African striped mice, wild house mice, and deer mice [12, 27, 62, 63].

**(C) Dark-light elevated platform test (DL/EP).** At the next stage, the cylinder was removed from the arena, the gerbil was returned to the box, and the box was placed on the elevated (16 cm) Plexiglas transparent platform (50 x 8 cm) with no walls located at the short side of the arena and directed toward the center. The box occupied half of the platform representing the "closed arm" of the apparatus, and its door looked toward the "open arm". After 1 min of habituation, the door was opened distantly, and the latencies to the "head out" and "body out" events were recorded. Then the time it took the gerbil to climb down (CD) was measured.

If an animal did not climb down within 5 min after the start of the experiment, the latency was set to 300 s. This test was adapted from the elevated plus-maze tests used to study anxiety-like and fear-like behaviour in laboratory and wild rodents [12, 64].

**(D) Dark-light stranger test (DL/STR).** At the final stage, the platform was removed, and the gerbil was returned to the box placed in the arena in the same way as in the Dark-light/startle/novel object test. A wire-mesh cage (15 x 6.5 x 6.5 cm) with an unfamiliar adult male was placed at the far side, opposite the box with a focal gerbil. After 1 min of habituation, the door was opened distantly. The focal animal could see the stranger in the cage. The latencies to the "head out" and "body out" were recorded. Then the time it took the gerbil to approach and contact (sniff) the stranger through the wire mesh (CS) was measured. If an animal did not approach a stranger during the test period of 5 min, the latency was set to 300 s. A similar "conspecific test" was used to measure social shyness in mink [65].

Thus, three composite tests began in the same way with a dark-light emergence assay and included two same behavioural responses ("head out" and "body out") as measures of boldness but differed in the third, final measurement—the latency to the final event (FE), which was test-specific and assessed different behaviours (Table 1, Fig 1). The latency to contact the novel object (CO) measured exploration, the latency to climb down (CD) from the platform measured anxiety (fear-related behaviour), and the latency to contact a stranger (CS) measured sociability. The total duration of each test in the arena, from the start to the final event, was also measured, and the times were summed up to estimate the total time the gerbils needed to complete a trial ("trial execution time"–ET). We used this measure as an estimate for the overall trial execution speed. At the end of the trial, the setup and equipment were cleaned with 70% Ethanol, and the floor of the arena was covered with fresh sand before introducing the next animal.

**Ethical note.** All applicable international and national guidelines for the care and use of animals were followed. All procedures conform to the ASAB/ABS Guidelines ethical treatment of animals [66]. The research protocol for this study was approved by the Animal Ethics Commission of Severtsov Institute of Ecology and Evolution (protocol #57–2022).

## Data analysis

**The consistency of behavioural responses across contexts.** We tested for cross-context consistency of individual behavioural responses by correlating the same single measurements of boldness (the latencies to "head out" and "body out" from the dark-light emergence assay) across three composite tests (dark-light/startle/novel object test, dark-light elevated platform test, and dark-light stranger test). We estimated pairwise correlations between same measurements of boldness in two different contexts (e.g. "head out" in the startle/novel object test and stranger test, or "body out" in the elevated platform and stranger tests) using bivariate Bayesian generalized linear mixed-effect models with Markov chain Monte Carlo (MCMC) techniques implemented in the R package MCMCglmm [67, 68]. Bayesian approach is a good solution for issues due to small sample sizes [69, 70]. Models included a pair of same measurements of boldness in two different contexts as the dependent variable and animal ID as a random effect allowing the estimation of both among-individual ($V_{ind}$) and residual (i.e. within-individual, $V_R$) variances of each measurement, as well as covariance between the two measurements. Phenotypic correlations were calculated as covariance divided by the square root of the product of the variances. We used slightly informative inverse-Wishart priors (R structure degree of belief (nu) = 0.002; G structure degree of belief (nu) = 0.004) with Gaussian distribution, which can produce valid and reasonable results even with small sample sizes [70–73]. Posterior distributions were based on 650,000 iterations with a burn-in period of 150,000 iterations and

a thinning interval of 1000 and ensured good model convergence yielding 500 iterations to obtain correlation estimates along with 95% credible intervals (CRIs) measured as the 95% highest posterior density (HPD). Credible intervals that do not overlap zero indicate statistical significance.

**The consistency (repeatability) of behavioural responses over time.** To assess the temporal consistency of behavioural responses, we used Bayesian regression analyses with the non-informative reference prior implemented in BAS R package [74, 75] to test whether the behaviour of an individual in the first trial predicts its behaviour in the second trials. We used trial execution time (ET) as well as composite variables obtained with principal component analysis (PCA) from a set of original behavioural measurements, a common and valid tool to reduce the dimensionality of multicollinear datasets in personality research [29, 76]. PCA was conducted with the complete data set of primary measurements (Table 1), which combined data from both species. Of 10 primary measurements, one ("head out" in the dark-light stranger test) was excluded from the PCA as highly correlated ($r > 0.85$) with "body out" in the same test. PCs were interpreted as composite behavioural responses according to the loadings of primary variables ($> 0.6$ in absolute value as recommended for small sample sizes–[76]). Each animal was assigned scores from each of the retained PCs. ET and PC scores in the first trial were used as predictors for ET and PC scores in the second trial.

In addition, to determine the proportion of the total behavioural variation attributable to between-individual variation [77], we estimated the coefficient of repeatability, $R$, for each of the three PCs and ET based on the univariate linear mixed-effect models (LMM) for Gaussian distribution with animal ID as a random factor using *rpt* function from the package rptR [78] in R 4.2.3 [79]. Confidence intervals (CIs, 95%) around $R$ and *p*-values based on the likelihood ratio test (LRT) were estimated via 1,000 bootstrapping runs.

**Correlation between different behaviours.** To assess whether different behaviours covary within individuals as a syndrome, we tested if they are correlated using bivariate Bayesian generalized linear mixed-effect models in the R package MCMCglmm with an animal ID included as random factor [67, 68]. In particular, we investigated the pairwise relationships between docility (immobility in the bag), boldness (the latency to show full body (BO) in the startle test), exploration (the latency to contact the novel object (CO)), anxiety (the latency to climb down (CD) from the platform), and sociability (the latency to contact a stranger (CS)). Note that boldness, exploration, and sociability were measured with the opposite to the traits metrics—the latencies to events (BO, CO, and CS, respectively), so that the shorter the latencies, the bolder, more explorative, and more sociable the individuals are. Docility and anxiety were measured with the direct metrics—the more time immobile and the longer the latency to CD, the more docile and anxious the individual, respectively.

**Univariate models for fixed and random (animal ID) effects.** We used univariate mixed-effects models—the best approach currently available in personality research [4]–to assess the variation in behavioural traits between species, sexes, and individuals and test the effect of the trial number. Significant effect of the trial number would indicate habituation, i.e. the change of individual response in time, and should be controlled in personality studies [24, 35], while the effect of the animal ID would indicate non-independence of individual responses, i.e. consistent inter-individual differences in behaviours [77]. We used the R package MCMCglmm [67, 68] to run four separate univariate Bayesian linear mixed-effect models with Markov chain Monte Carlo estimation for each of the three PCs and the trial execution time with species, sex, and the trial number (as a two-level factor) included as fixed effects and animal ID as a random factor.

Since we had two repeated individual measurements, we could partition behavioural variance into between-individual ($V_{ind}$) and residual ($V_R$) variances [80]. We used slightly

informative inverse-Wishart priors (R structure degree of belief (nu) = 0.002; G structure degree of belief (nu) = 0.004) with Gaussian distribution and variance equal to $10^3$. As recommended [72, 73], to ensure proper model convergence, we adjusted priors (1) based on previous studies with comparable sample sizes and test procedures (e.g. [12]) and (2) by checking chains (five separate chains per model) for autocorrelation and the posterior distributions. Initial models showed some autocorrelation, and finally we ran the models with 650,000 iterations, a burn-in period of 150,000, and a thinning interval of 500 yielding Monte Carlo Markov chains with a sample size of 1,000 to obtain point estimates and 95% CRIs. Chains exhibited low autocorrelation (less than 0.1), and distributions were unimodal, smoothed, and with no gaps, which is indicative of sufficient effective sample size and proper model convergence [69, 72]. We report the posterior mean as a point estimate along with the 95% CRIs measured as the 95% HPD for fixed effects. We report posterior modes with 95% HDP CRIs as estimates of between-individual variance component ($V_{ind}$) for the random effect of animal ID.

Statistical analysis was performed in R 4.2.3 [79]. All measurements prior to analyses were ln($x$+1) transformed to improve normality. Data are presented as raw data means and posterior means or modes from model predictions with 95% confidence intervals, if not specified otherwise.

## Results

### Consistency of behaviours across contexts

In *M. meridianus*, both measurements of boldness ("head out" and "body out" in dark-light emergence assay) were highly and significantly positively correlated between all three different contexts (startle/novel object, elevated platform, and stranger tests) (Table 2). In *M. unguiculatus*, only three of six possible correlations were significant, which is still more than expected by random (assuming $\alpha$ = 0.05). Phenotypic correlations in *M. meridianus* were mostly driven by among-individual correlations, whereas in *M. meridianus* mostly by within-individual correlations (S1 Table).

### Consistency of behaviours over time

PCA of primary behavioural measurements obtained in the first and second trials of each individual of both species in all four tests produced three PCs with eigenvalues > 1.0, which in

**Table 2. Correlations between the same measurements of boldness across three tests (contexts).**

| Test | *M. meridianus* | | *M. unguiculatus* | |
|---|---|---|---|---|
| | **DL/EP** | **DL/STR** | **DL/EP** | **DL/STR** |
| | Head out (HO) | | | |
| **DL/S/NO** | **0.66 (0.23; 0.82)** | **0.53 (0.23; 0.85)** | **0.71 (0.29; 0.93)** | 0.34 (-0.14; 0.80) |
| **DL/EP** | | **0.69 (0.38; 0.90)** | | **0.75 (0.49; 0.96)** |
| | Body out (BO) | | | |
| **DL/S/NO** | **0.79 (0.52; 0.92)** | **0.82 (0.61; 0.95)** | <u>0.51 (-0.01; 0.88)</u> | 0.54 (-0.09; 0.80) |
| **DL/EP** | | **0.65 (0.15; 0.80)** | | **0.91 (0.67; 0.98)** |

Correlations were derived from the bivariate MCMCglmm models with animal ID included as random factor. Estimates along with 95% CRIs (in brackets) were calculated as covariance divided by the square root of the product of the variances. Significant results (0 not included in the 95% CRIs) are marked with bold font; marginal significant correlations are marked with underlined font. DL—the dark-light-emergence stage of the tests: S/NO—startle/novel object, EP—elevated platform, STR—stranger tests.

**Table 3. Results of the PCA (PCA loadings) of primary behavioural measurements.**

| Test | Measurements | PC 1 | PC 2 | PC 3 |
|---|---|---|---|---|
| Bag | Immobility | -0.37 | **0.76** | 0.15 |
| DL/S/NO | HO | **-0.81** | -0.26 | -0.25 |
| | BO | **-0.87** | -0.14 | -0.06 |
| | CO | -0.53 | 0.08 | **-0.68** |
| DL/EP | HO | **-0.82** | -0.04 | -0.15 |
| | BO | **-0.77** | -0.32 | 0.21 |
| | CD | -0.39 | -0.32 | **0.66** |
| DL/STR | BO | **-0.82** | 0.20 | 0.30 |
| | CS | -0.35 | **0.73** | 0.11 |
| Eigenvalue | | 4.0 | 1.5 | 1.2 |
| % Total variance | | 44.9 | 16.1 | 12.9 |
| PC label | | Boldness | Docility/ Social shyness | Exploration/ Anxiety |

PCA loadings > 0.6 in absolute value [76] are in bold font. HO in DL/STR test was excluded as highly correlated ($r > 0.85$) with BO in the same test.

DL—the dark-light-emergence stage of the tests: S/NO—startle/novel object, EP—elevated platform, STR—stranger tests.

HO—head out, BO—body out, CO—contact object, CD—climb down, CS—contact stranger. Note that Boldness, Exploration, and Sociability were measured with the opposite to the traits metrics—the latencies to events (HO and BO, CO, and CS, respectively), so that the shorter the latencies, the bolder, more explorative, and more sociable the individuals are. Docility and Anxiety were measured with the direct metrics—the more time immobile and the longer the latency to CD, the more docile and anxious the individual, respectively. Therefore, the negative (positive) loadings of the relevant primary variables (HO and BO, CO, and CS) indicate positive (negative) correlations with respectively boldness, exploration, and sociability. The signs of loadings for immobility and CD indicate the direction of correlation with docility and anxiety, respectively.

total explained 74% of the variance (Table 3). PC1 was associated with short latencies to "head out" and "body out" in the dark-light emergence assays in all three contexts and was labelled "Boldness". PC2 was complex and reflected the low activity level in the bag, on the one hand, and the long latency to contact stranger, on the other. Accordingly, we interpreted it as a measure of "Docility/ Social shyness". PC3 combined the short latency to contact object in the startle/novel object test and the long time to climb down in the elevated platform test; we named it "Exploration/Anxiety".

Individual PC1 scores (Boldness) in the first trial reliably predicted the scores in the second trial in both species (Table 4, Fig 2a). PC2 scores (Docility/Social shyness) were significantly correlated between the first and second trials only in *M. meridianus* (Fig 2b). PC3 scores (Exploration/Anxiety) in the second trial depended on the scores in the first trial in *M. unguiculatus* but not in *M. meridianus* (Fig 2c). ET was positively related between repeated trials in both species (Fig 2d), so that the faster gerbil was in the first trial, the faster it was in the second one.

Behavioural repeatability estimates obtained from the univariate LMMs with animal ID as a random factor supported the results (Table 5). In *M. meridianus*, Boldness and Docility/Social shyness were highly repeatable with most of the total variance explained by between-individual differences. *M. unguiculatus* exhibited high repeatability in Boldness and Exploration/Anxiety. The repeatability of the trial execution time was high in both species.

## Correlations across different behaviours

In *M. meridianus*, among ten possible correlations between different behaviours, only two were significant. Docility (immobility in the bag) was related positively to the latency to

**Table 4. Regression analysis of individual behavioural responses in the first and second trials.**

| Species | $R^2$ | Intercept | $\beta$ |
|---|---|---|---|
| | | PC1 –Boldness | |
| *M.m.* | **0.65** | -0.08 (-1.13; 0.95) | **1.06 (0.49; 1.65)** |
| *M.u.* | **0.82** | **0.64 (0.08; 1.17)** | **0.91 (0.52; 1.27)** |
| | | PC2 –Docility/Social shyness | |
| *M.m.* | **0.59** | **-0.57 (-1.13; -0.01)** | **0.93 (0.38; 1.55)** |
| *M.u.* | 0.16 | **0.79 (0.27; 1.27)** | 0.12 (-0.17; 0.69) |
| | | PC3 –Exploration/Anxiety | |
| *M.m.* | 0.00 | 0.06 (-0.67; 0.74) | -0.02 (-0.64; 0.49) |
| *M.u.* | **0.63** | -0.09 (-0.75; 0.61) | **0.84 (0.00; 1.36)** |
| | | ET | |
| *M.m.* | **0.37** | **5.7 (5.3; 6.1)** | **0.58 (0.00; 1.26)** |
| *M.u.* | **0.89** | **4.6 (4.3; 4.9)** | **1.03 (0.71; 1.36)** |

Coefficients of the Bayesian regression along with 95% CRIs (in brackets) for individual PC scores and the trial execution time (ET) in the second trial depending on the scores in the first trial. Significant coefficients (0 not included in the 95% CRIs) are marked with bold font. *M.m.–M. meridianus*, *M.u.–M. unguiculatus*.

contact stranger (i.e. social shyness): more docile animals were less sociable (Table 6). The latency to "body out" in the dark-light startle test was correlated positively with the latency to climb down in elevated platform test, i.e. bolder individuals exhibited lower anxiety. Also, bolder individuals tended to be faster explorers as indicated by the positive correlation close to

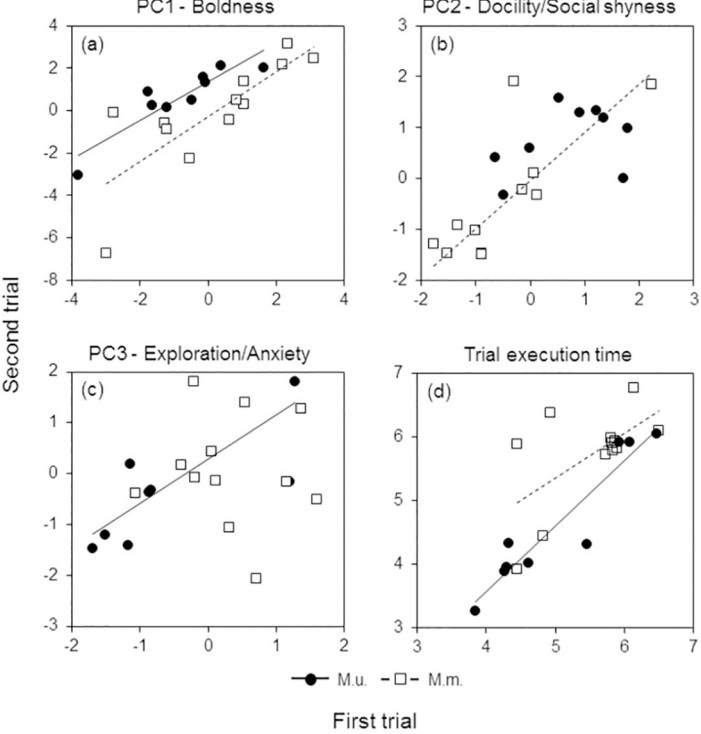

**Fig 2. Behavioural responses in the second trial depending on the responses in the first trial.** (a) PC1 –Boldness, (b) PC2 –Docility/Social shyness, (c) PC3 –Exploration/Anxiety (as in Table 3), and (d) the trial execution time. M.m.–*M. meridianus*, M.u.–*M. unguiculatus*.

**Table 5. Repeatability of behavioural responses.**

| Behavioural traits | M. meridianus | | M. unguiculatus | |
|---|---|---|---|---|
| | Estimate (CI) | p | Estimate (CI) | p |
| PC1 Boldness | **0.78 (0.40; 0.93)** | **0.0006** | **0.58 (0.01; 0.88)** | **0.04** |
| PC2 Docility/Social shyness | **0.77 (0.36; 0.93)** | **0.0008** | 0.42 (0.00; 0.81) | 0.12 |
| PC3 Exploration/Anxiety | 0.002 (0.00; 0.55) | 0.5 | **0.77 (0.27; 0.94)** | **0.003** |
| ET | **0.60 (0.081; 0.85)** | **0.01** | **0.87 (0.50; 0.97)** | **0.0003** |

Repeatability estimates along with their 95% CIs (in brackets) for behavioural traits (three PCs as in Table 3, and the trial execution time—ET) obtained from the univariate LMMs in rptR package with animal ID as random factor. *p*- values were estimated with the likelihood ratio test (LRT). Significant coefficients are marked with bold font.

significance between the latency to "body out" in the dark-light startle test and the latency to contact object in the novel object test. In *M. unguiculatus*, the latency to "body out" in the dark-light startle test was positively related to the latency to contact stranger, i.e. the animals that were bolder when startled, were more sociable. Also, more sociable Mongolian gerbils showed less anxiety in elevated platform test as indicated by the positive correlation between the latency to contact stranger and the latency to climb down. All other pair-wise correlations were weak or moderate at the best and far from significance. Phenotypic correlations were mostly driven by among-individual correlations (S1 Table).

**Table 6. Phenotypic correlations between different behaviours.**

| Test | Measurements | Behavioural traits | BO Boldness | CO Exploration | CD Anxiety | CS Sociability |
|---|---|---|---|---|---|---|
| | | | *M. meridianus* | | | |
| Bag | Immobility | Docility | 0.42 (-0.08; 0.81) | 0.14 (-0.21; 0.59) | 0.24 (-0.30; 0.63) | **0.51 (0.14; 0.82)** |
| DL/S/NO | BO | Boldness | | 0.46 (-0.01; 0.76) | **0.30 (0.01; 0.78)** | 0.02 (-0.38; 0.55) |
| DL/S/NO | CO | Exploration | | | 0.10 (-0.30; 0.50) | 0.45 (-0.05; 0.70) |
| DL/EP | CD | Anxiety | | | | 0.17 (-0.29; 0.58) |
| | | | *M. unguiculatus* | | | |
| Bag | Immobility | Docility | 0.06 (-0.48; 0.52) | -0.12 (-0.63; 0.46) | 0.38 (-0.08; 0.87) | 0.33 (-0.24; 0.68) |
| DL/S/NO | BO | Boldness | | 0.48 (-0.10; 0.74) | 0.22 (-0.24; 0.71) | **0.62 (0.16; 0.87)** |
| DL/S/NO | CO | Exploration | | | -0.29 (-0.83; 0.23) | 0.13 (-0.39; 0.62) |
| DL/EP | CD | Anxiety | | | | **0.59 (0.01; 0.86)** |

Correlations were derived from the bivariate MCMCglmm models with animal ID included as random factor. Estimates along with 95% CRIs (in brackets) were calculated as covariance divided by the square root of the product of the variances. Significant results (0 not included in the 95% CRIs) are marked with bold font; marginal significant correlations are marked with underlined font.

DL—the dark-light-emergence stage of the tests: S/NO—startle/novel object, EP—elevated platform, STR—stranger tests.

BO—body out, CO—contact object, CD—climb down, CS—contact stranger. Note that Boldness, Exploration, and Sociability were measured with the opposite to the traits metrics—the latencies to events (BO, CO, and CS, respectively), so that the shorter the latencies, the bolder, more explorative, and more sociable the individuals are. Docility and Anxiety were measured with the direct metrics—the more time immobile and the longer the latency to CD, the more docile and anxious the individual, respectively. Therefore, the positive correlation coefficients between docility and sociability and between boldness and anxiety in *M. meridianus* indicate that more docile individuals were less sociable and more bold individuals were less anxious, respectively. Likewise, in *M. unguiculatus*, the positive coefficient of correlation between anxiety and sociability indicates that more anxious individuals were less sociable, whereas the positive correlation between boldness and sociability indicates that bolder individuals were more sociable.

**Table 7. Effects of species, sex, trial number (fixed effects), and animal ID (random factor) on behavioural responses in gerbils.**

| Effects | Behavioural traits | | | |
|---|---|---|---|---|
| | PC1<br>Boldness | PC2<br>Docility/<br>Social shyness | PC3<br>Exploration/<br>Anxiety | ET |
| **Fixed effects** | Estimate<br>(95% CRI) | Estimate<br>(95% CRI) | Estimate<br>(95% CRI) | Estimate<br>(95% CRI) |
| **Intercept** | 0.51<br>(-1.28; 2.18) | -0.54<br>(-1.50; 0.33) | 0.47<br>(-0.52; 1.49) | **5.67<br>(4.93; 6.45)** |
| **Species(mu)** | -1.05<br>(-3.09; 0.82) | **1.25<br>(0.04; 2.37)** | -0.81<br>(-2.00; 0.45) | -0.52<br>(-1.38; 0.41) |
| **Sex(male)** | -0.54<br>(-2.31; 1.38) | -0.02<br>(-0.96; 1.03) | -0.23<br>(-1.33; 0.72) | -0.27<br>(-1.12; 0.63) |
| **Trial (#2)** | -0.27<br>(-1.07; 0.51) | -0.04<br>(-0.42; 0.55) | -0.27<br>(-0.89; 0.44) | 0.21<br>(-0.09; 0.57) |
| **Species*Trial** | **1.72<br>(0.54; 2.87)** | 0.13<br>(-0.54; 0.87) | 0.63<br>(-0.49; 1.58) | **-0.60<br>(-1.09; -0.12)** |
| **Random effect** | | | | |
| **Animal ID** | **2.98<br>(1.46; 6.90)** | **0.95<br>(0.44; 2.29)** | **0.75<br>(0.29; 1.92)** | **0.82<br>(0.41; 1.81)** |

Posterior estimates along with 95% CRIs were derived from MCMC linear mixed-effect models run separately for each of the three PC and the trial execution time—ET. Point estimates show posterior means for fixed effects and posterior modes for between-individual variance (animal ID) along with their 95% credible intervals measured as the 95% highest posterior density (HPD). *M. unguiculatus* (mu), male, and trial #2 were considered as the references for species, sex, and trial# in the models, respectively. Significant effects are marked with bold font.

## Predictors of behavioural responses

Univariate MCMC LMMs run separately for each of the three PCs and the trial execution time showed that the species effect was significant for PC2 (Docility/Social shyness) with higher scores in *M. unguiculatus* than in *M. meridianus* (Table 7, Fig 3b). No effect of sex was observed for any of the three PCs or the trial execution time (Fig 3). The trial number significantly influenced PC1 (Boldness) and the trial execution time in the interaction with species: in *M. unguiculatus*, estimates for PC1 were higher, whereas estimates for ET were lower in the second trial than in the first trial (raw-data mean ± 95%CI for PC1 scores: 0.64 ± 1.20 vs -0.80 ± 1.19, respectively; ln-transformed ET: 4.63 ± 0.81 vs 5.03 ± 0.74, respectively), i.e. *M. unguiculatus* were bolder and faster in the second trial as compared to the first trial. In *M. meridianus*, none of the three PC scores or ET varied between trials. The random effect of animal ID was highly significant for all three PCs and ET, suggesting that repeated individual measurements were not independent and that there were consistent inter-individual differences in Boldness, Docility/Social shyness, Exploration/Anxiety, and trial execution speed in both species.

## Discussion

Using a suit of multivariate repeated behavioural tests, we found contextually consistent and highly repeatable sex-independent but species-specific personality traits in the solitary midday gerbil, *M. meridianus*, and social Mongolian gerbil, *M. unguiculatus*. Species differed in the temporal repeatability of different behaviours as well as contextual consistency, which was more pronounced in solitary *M. meridianus* than in social *M. unguiculatus*. The habituation effect indicative of learning abilities was weak in both species yet stronger in social *M.*

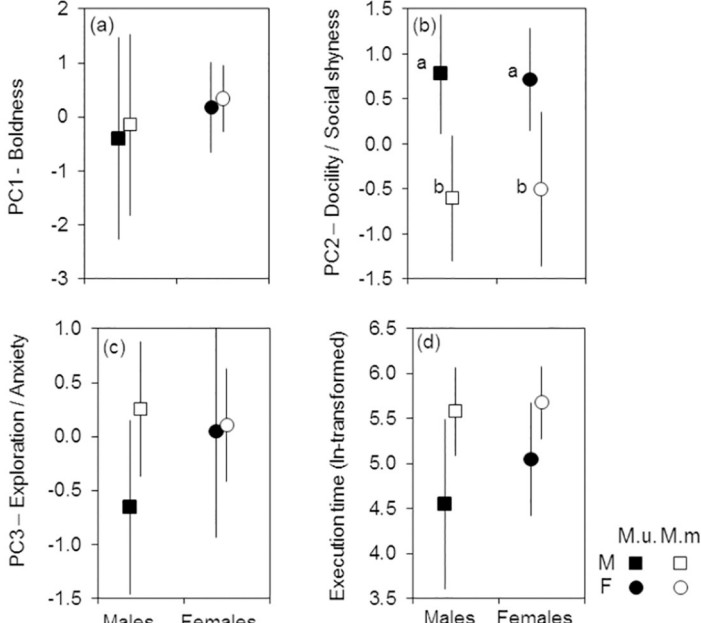

**Fig 3. Between-sex and between-species variation in behavioural responses of gerbils.** (a) PC1 scores—Boldness, (b) PC2 scores—Docility/Social shyness, (c) PC3 scores—Exploration/Anxiety (see Table 3), and (d) the trial execution time (ln-transformed). Means ± 95%CIs from raw data. M.u.–*M. unguiculatus*, M.m.–*M. meridianus*.

*unguiculatus*. In both species, only a few different behavioural traits covaried, and the sets of correlated behaviours were species-specific such that the two species did not share any pair of correlated traits.

## The utility of the test procedure

Our procedure meets most requirements for the personality tests: (1) it includes a combination of the same as well as different multiple individual behavioural measurements (2) estimated in different but overlapping spatial and temporal contexts (i.e. measured in the same experimental compartment in sequential assays) and (3) repeated over time [4, 8, 31, 33, 34]. This allows testing for the consistency of multivariate among-individual differences in behaviour across contexts and over time—a key characteristic of personality. In addition, it does not require sophisticated equipment or setup design and is not time-consuming (mean ± SD procedure duration was 15 ± 3 min). This rigorous but simple, fast, and convenient procedure can be used not only in the laboratory but also under field conditions in a variety of small mammal species, which is important to advance comparative studies of animal personality [29].

## Consistency of behaviours across contexts

The contextual consistency of behaviour is a key characteristic of personality [2]; however, it is not always tested in personality studies [10, 35]. In our study, the measures of boldness at emergence from the shelter (the latencies to "head out" and "body out") were positively and highly correlated among different contexts (Table 2) in *M. meridianus* and to a lesser degree in *M. unguiculatus*, which was more flexible and context-specific in behavioural responses. Thus, in both species of gerbils, the individual behavioural responses exhibited contextual consistency, indicating personality traits more pronounced in solitary *M. meridianus* and less

prominent in social *M. unguiculatus*. High contextual consistency was previously shown for *M. meridianus* in sociability and partner preference, which remained consistent across different test designs, unlike *M. unguiculatus* [52].

Not in all personality studies the same behavioural trait was found to be consistent across contexts [34, 35]. The problem arises from measuring different behaviours to assess "the same" behavioural trait [3]. For example, in damselfish, the measure of boldness in the emergence test was not correlated with boldness measured as activity outside refuge, which made authors question the consistency of boldness as a personality trait [35]. Personality traits are often interpreted too broadly as an umbrella term for distinct behaviours driven by context-specific different motivational processes [34]. In our study, each of the three composite tests began with the same dark-light emergence assay conducted in different contexts (startle, platform, and stranger tests) (Fig 1). Thus, we did measure the same behaviour, which can be interpreted as a "boldness/shyness" individual trait, and showed its contextual species-specific consistency higher in solitary *M. meridianus* and lower in social *M. unguiculatus*.

## Consistency (repeatability) of behaviours over time

Individual behavioural traits were repeatable over time to a large degree in both gerbil species, although in different ways. Among-individual variation in boldness and the trial execution speed was highly consistent over time in both *M. meridianus* and *M. unguiculatus* (Fig 2a and 2d, Tables 4 and 5). Bolder and faster individuals in the first trial were also bolder and faster in the second trial. However, docility/social shyness was repeatable only in *M. meridianus* (Fig 2b), whereas exploration/anxiety only in *M. unguiculatus* (Fig 2c). We suggest that more consistent among-individual variation in social shyness in *M. meridianus* may be attributed to social indifference, the key life-history feature of this species [51, 54]. Social indifference implies living in the lowly variable perceived social environment, which, in turn, may favour more stereotypic and more consistent behaviour according to the hypothesis that behavioural plasticity increases with environmental variability [81]. In other words, behavioural responses of *M. meridianus* in the social context might not be influenced situationally by relatively neutral to the subject social stimulus but rather reflected inherent and context-independent individual sociability, less flexible and thus more stable than in social *M. unguiculatus*. Between-species differences in temporal plasticity of behaviour in the social context (high in social *M. unguiculatus* and low in solitary *M. meridianus*) are consistent with higher contextual flexibility in sociability and partner preferences found in *M. unguiculatus* in the previous study [52]. Overall, both species demonstrated high temporal consistency (repeatability) of individual behavioural responses, another key characteristic of personality, but differed in trait-specific plasticity–*M. unguiculatus* was more flexible in social context whereas *M. meridianus* in non-social contexts.

## Correlations across different behaviours

Different behaviours were generally not correlated in either gerbil species, with few exceptions, and the pairs of correlated behaviours were species-specific such that none overlapped between species (Table 6). In *M. meridianus*, docility was positively related to social shyness, while boldness correlated negatively with anxiety (fear-related behaviour) and tended to correlate positively with exploration. In *M. unguiculatus*, higher boldness was associated with higher sociability, which in turn was negatively related to anxiety. The higher boldness of more sociable animals found in *M. unguiculatus* is consistent with some studies but contradicts others. For example, boldness correlated positively with sociability in American mink [65] and negatively in the three-spined stickleback, *Gasterosteus aculeatus* [7]. Given that Mongolian gerbils, unlike *M. meridianus*, commonly exhibit stranger-directed aggression both in the field [56, 57]

and in the laboratory tests [52], our measure of sociability (the latency to contact a stranger) characterizes social boldness rather than sociability *per se* in this species. In other words, Mongolian gerbils that were bolder in the non-social contexts (in the startle test and the elevated platform test) were also bolder in the social context (Table 6) constituting a personality bold in various aspects. The difference between gerbil species in stranger-directed behaviour also suggests that the same metric may reflect similar but different behavioural traits (e.g. social affiliation or social boldness in conspecific tests) depending on species specificity, which should be considered when interpreting the results of personality tests [8].

The negative correlation between measurements of boldness and anxiety (fear-like behaviour) in *M. meridianus* implies that different tests we used (emergence/startle test and elevated platform test) measuring different behaviours assessed the same risk-taking personality trait in this species. Boldness-exploration syndrome found in solitary *M. meridianus* supports other similar findings [1, 82–84, but see 23], whereas the lack of such correlation in social *M. unguiculatus* contradicts them. These controversial results, which may be attributed to species specificity, are consistent with the findings by Wey et al. [27] for *Peromyscus* species. They revealed boldness-exploration syndrome only in non-monogamous, less social deer mice but not in monogamous and social species.

We did not find an association between docility and boldness, anxiety or exploration in either species—only social shyness and only in *M. meridianus* was positively correlated with docility. Similarly, docility did not covary with boldness in *Marmota flaviventris* [85] or with exploration in deer mice [27] and Belding's ground squirrels, *Urocitellus beldingi*, [26], nor with activity/exploration, reaction to stress, or emotionality in chipmunks, *Tamias striatus*, [24] suggesting its independence of other traits [85]. Nevertheless, in contrast to these findings, in other studies, docility was correlated negatively with exploration and boldness [23, 83, 86]. Finally, we did not find sociability to be correlated with exploration in either gerbil species, contrary to the negative association between these behavioural traits in the common waxbill, *Estrilda astrild*, [87] and deer mice [27].

Such equivocal and inconsistent results, common for personality studies, may be due to different measures of similarly labelled behaviours [3] or species specificity of behavioural responses. For example, in a comparative study, a syndrome between activity and agonistic behaviour was found in only one of four shrew species [43]. Likewise, in deer mice [27] and gerbils in our study, behavioural syndromes differed between closely related species. All this questions the existence of universal behavioural syndromes and cautiously suggests that behavioural correlations are likely species-specific depending on the species' life history and social system. However, the lack of comparative studies under standardized conditions does not allow making strong inferences.

Overall, our results only partly support and mostly contradict the behavioural syndrome concept [1, 5, 88]. This suggests that in gerbils, different personality traits are mainly context-specific, underpinned by distinct motivational processes, and not governed by some common individual background trait, e.g., temperament. The relative independence of different behaviours in gerbils is consistent with the syndrome deviation concept [89], which assumes relaxed constraints for the coupled evolution of personality traits. Under this view, the lack of a behavioural syndrome may be advantageous in a changing environment by providing high individual behavioural plasticity.

## Effects of sex, animal ID, and trial

The best and the only significant predictor of all four behavioural traits (boldness, docility/social shyness, exploration/anxiety, and trial execution speed) was an animal's ID, which

explained a major proportion of the total behavioural variance (Tables 5 and 7). The effect of individual identity overwhelmed all other effects, supporting our conclusions on the existence of personality traits in both species of gerbils. Previously, consistent among-individual differences in boldness and exploration were found in *M. unguiculatus* [25].

The sex of an individual did not influence either behaviour, and behavioural traits varied between sexes less than between individuals, consistent with many other studies (*Microtus arvalis*–[6], *Neovison vison*–[65], *Mus musculus*–[39], but see [28]). In *Peromyscus*, only non-monogamous species demonstrated sexual dimorphism and only in sociability but not in boldness, activity, or exploration [27]. The lack of prominent sex differences in gerbils supports the general conclusion of the recent meta-analysis that personality is not a sex-specific feature [90].

The trial number influenced only boldness and the trial execution speed and only in *M. unguiculatus*: Mongolian gerbils were bolder and faster in the second trial, indicating the habituation to the fearful situation [24]. Habituation of *M. unguiculatus* to fearful and novel situations was also shown previously [25]. *M. meridianus* showed no changes in behaviours in the repeated trial. Habituation effect may vary between species [91], sexes [65], or individuals [35, but see 24] and indicates some simple form of learning and cognitive abilities [92]. Our results show habituation to be low in both species yet higher in social *M. unguiculatus* than in solitary *M. meridianus*, which supports the relationship between the sociality level and cognitive skills [93, 94].

## Between-species differences

Comparative studies show that behavioural consistency may be trait-, context-, and(or) species-specific [27, 29, 40, 43]. Gerbil species differed in only one behaviour—docility/social shyness, with *M. unguiculatus* being more docile and shy in the contacts with a stranger, which corresponds to species-specific social behaviour: Mongolian gerbils, unlike midday gerbils, exhibit individual and group territoriality and stranger-directed aggression [56, 57]. Thus, they are likely to be more cautious when encountering a stranger. This is consistent with previous findings–*M. meridianus* showed no stranger-directed aggression or social preference between familiar partner and stranger in preference tests, exhibiting "social indifference", whereas *M. unguiculatus* preferred partners and avoided strangers [52].

The main differences between species were observed in the patterns of behavioural consistency. We attributed the between-species variation in habituation and behavioural repeatability to the species-specific social behaviour (see above). Likewise, different contextual consistency of behaviours may be related to the species' sociality. The social niche specialization hypothesis suggests that personality traits should be more prominent in more social species because the diversity of consistent behavioural types reduces social competition within groups and, thus, should be favoured [42]. However, this theoretical prediction, so far, has been tested in only a few comparative studies of closely related but socially different species [44]. The only two available studies for mammals produced equivocal results. In deer mice, the hypothesis was not supported as more social species did not show more pronounced behavioural correlations than less social species [27]. In shrews, consistent with predictions, personality traits were prominent in social *Neomys anomalus* but not in solitary species [43]. We found the opposite tendency—contextual consistency was more pronounced in solitary *M. meridianus*, which contradicts the social niche specialization hypothesis. Alternatively, we suggest that higher behavioural plasticity (in particular, in social shyness) in social *M. unguiculatus* compared to solitary *M. meridianus* is determined proximately by higher learning abilities (manifested by higher habituation) and ultimately by living in a more complex social

environment. Under this "the social behaviour flexibility" hypothesis, learning ability provides whereas social complexity requires more flexibility and therefore less consistency in behavioural responses in social than solitary species. This corresponds to the idea that a more variable environment (social, in case of *M. unguiculatus*) selects for higher cognitive abilities and behavioural flexibility [81].

Different origins of gerbils (wild-caught *M. meridianus* vs captive-bred *M. unguiculatus*) could have potentially biased the results [95]; however, this is unlikely to apply to our study. First, the young of both species were reared after weaning under the same conditions, which could unlikely influence the development of behavioural traits in different ways. Second, domestication or any other unintentional selection could have hardly occurred during 14 years in the outbred zoo colony where life span of gerbils was about two years. Wild rodents from outbred colonies kept in captivity for decades were successfully used to investigate species-typical behaviour in a standardized environment [27]. Then, Mongolian gerbils in the zoo colony were kept in pairs in large cages (100 x 40 x 40 cm) with the enriched environment and shelters [52] and were trapped and examined shortly only for litter separation or breeding pair formation, which should weaken nonintentional selection on domestication. Further, gerbils had no white spots in the fur and did not demonstrate stereotypic "flipping" or "gnawing" behaviours indicative of the domestication syndrome in rodents [96, 97]. Finally, and most importantly, domesticated animals are generally characterized by less boldness and exploration, more anxiety-like behaviour, and higher sociability compared to wild animals [98]. We did not find differences between the captive-bred Mongolian and wild-caught midday gerbils in boldness, exploration, or anxiety. Moreover, Mongolian gerbils were less sociable in their contacts with a stranger, which contradicts the "domestication hypothesis" but is consistent with species-typical behaviour demonstrated in the wild. In addition, Mongolian gerbils were more flexible in behavioural responses, whereas domesticated animals are expected to be less flexible [99]. Altogether, the different patterns of personality found in Mongolian and midday gerbils likely reflect differences in the species-typical behaviour rather than in the origin of animals.

## Small sample size issue

A relatively small sample size (12 *M. meridianus* and 9 *M. unguiculatus*, 21 individuals in sum) might have limited the power to detect the effects when the effect sizes were small [60]. Nevertheless, we believe the small sample size has not significantly influenced the results and main inferences. First, we tested gerbils under highly standardized experimental conditions and overlapping spatial and temporal contexts which keeps the majority of confounding factors constant. Standardized experimental conditions combined with repeated tests decrease the variance, reducing the required sample size, which is typically much smaller in experimental studies in captivity compared to the observational surveys in the field [60]. Second, there is a trade-off between sample size and effect size: a larger effect size allows for a smaller sample size and vice versa [60]. The significant results in our study showed medium or (mostly) strong effects (Tables 2, 4–7). Third, the statistical analysis was adjusted to small sample size, and we used Bayesian methods—a good solution for issues due to small sample sizes [69, 70]–which produced posterior distributions with the effective sample sizes of 500–1000. All in all, this suggests that sample size and statistical power were sufficient to detect medium and strong effects. Moreover, many non-significant results were associated with very small effects (e.g., Figs 2 and 3; Tables 4–7), suggesting a low probability of false negatives; nevertheless, the inference of "no effect" in those cases should be taken with caution [8].

## Conclusion

Using a simple multi-assay test procedure, we have revealed sex-independent species-specific personalities in closely related but socially distinct species of gerbils. Species differed in temporal and contextual consistency of traits, habituation, and the sets of correlated behaviours. Personality traits were more consistent in solitary species and were more flexible in social species. Together with previous findings [27, 52], this contradicts the social niche specialization hypothesis [42] tested so far in a couple of studies, calling for further comparative research on the links between personality and sociality. We failed to identify prominent behavioural syndromes—only a few behaviours covaried, and the sets of correlated behaviours were species-specific. These findings question the existence of universal behavioural syndromes, consistent with the idea that context-specific individual behavioural traits might be favoured to allow more flexible and adequate responses to the changing environment than syndromes of correlated functionally distinct and context-independent behaviours [40, 89].

## Supporting information

**S1 File. Dataset.** See Table 1 and Fig 1 for the description of behavioural measurements.
(CSV)

**S2 File. Codes.**
(DOCX)

**S1 Table. Correlations between behaviours.** See Table 1 and Fig 1 for the description of behavioural measurements.
(DOCX)

## Acknowledgments

We are grateful to A. Kalinin, T. Demidova, O. Batova, N. Shchipanov, and S. Popov for valuable comments on the test design and earlier versions of the paper. We thank A. Bogatchuk for assistance in conducting tests.

## Author Contributions

**Conceptualization:** Andrey V. Tchabovsky.

**Data curation:** Elena N. Surkova.

**Formal analysis:** Andrey V. Tchabovsky.

**Funding acquisition:** Andrey V. Tchabovsky.

**Investigation:** Andrey V. Tchabovsky, Elena N. Surkova, Ludmila E. Savinetskaya.

**Methodology:** Andrey V. Tchabovsky, Elena N. Surkova, Ludmila E. Savinetskaya.

**Project administration:** Andrey V. Tchabovsky.

**Resources:** Andrey V. Tchabovsky.

**Supervision:** Andrey V. Tchabovsky.

**Validation:** Andrey V. Tchabovsky.

**Visualization:** Andrey V. Tchabovsky.

**Writing – original draft:** Andrey V. Tchabovsky.

**Writing – review & editing:** Andrey V. Tchabovsky, Elena N. Surkova, Ludmila E. Savinetskaya.

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
