## [Decision Letter · Decision Letter 0]

8 Jan 2024

PONE-D-23-40950Multi-assay approach shows species-associated personality patterns in two socially distinct gerbil speciesPLOS ONE

Dear Dr. Tchabovsky,Thank you for submitting your manuscript to PLOS ONE. After careful consideration, we feel that it has merit but does not fully meet PLOS ONE’s publication criteria as it currently stands. Therefore, we invite you to submit a revised version of the manuscript that addresses the points raised during the review process.

We look forward to receiving your revised manuscript.

Kind regards,

Xiaona Wang

Academic Editor

PLOS ONE

Journal Requirements:

3. Thank you for stating the following financial disclosure:"This research was supported by the Russian Science Foundation (22-14-00223 to AVT, https://rscf.ru/project/22-14-00223/)"  

Reviewers' comments:

Reviewer #1

In the Introduction the personality and behavioural sindrome were thoroughly discussed and the cited references were more than enough to define the experimental problem adreesed by the research.

Although the dataset is quite small (12 and 9 individuals, respectively) the experimental procedure was correct, and described in details allowing thus the repeatibility of the experiment. Each survey was perfomed twice, at a 3-days interval, as suggested by literature. The M&M part is clear and the protocol is well-described. The chosen statistical analyses are sound and well-supported by the references.

The results are clearly shown, and no further analyses are required. In the Discussion the findings are assessed, and supported by the references. The alternative hypothesis suggested for the results contradicting the behavioural sindrome concept is acceptable.

The existence of personality traits in both species of gerbils is an interesting result, but the main concern of the present research is the small number of individuals included in the study, as also recognized by the authors (rows 674-696). The low numerosity could have affected the results, that shown some inconsistencies related to behavioural syndrome. Although the manuscript is overall interesting, the research could have been partly wrecked by the small dataset. I could suggest to include the part dealing with the issue of the small dataset in the M&M section, and then evaluate any potential effect on the present findings in the Discussion.

Reviewer #2

Having thoroughly reviewed the manuscript titled "Multi-assay approach shows species-associated personality patterns in two socially distinct gerbil species," I find no significant methodological difficulties or other issues that require immediate attention.

The articulation of objectives appears accurate and aligns well with the introduction, methods, and results. The interpretation of the findings not only prompts further investigation but also encourages a more in-depth analysis of various aspects related to animal personality and behavioral syndromes in small mammals, particularly rodents. To substantiate the social niche specialization hypothesis, additional studies are warranted, delving into the mechanisms of phenotypic integration and unraveling the evolutionary paths leading to the development of behavioral syndromes.

While the discussion section is comprehensive, I recommend a careful review to potentially condense its length. Certain parts are challenging to follow, and a more streamlined approach could enhance readability. Additionally, Figure S1 should be incorporated into the main text, as it significantly aids in comprehending the performed behavioral trails.

In the Materials and Methods section, consider minimizing the use of abbreviations (e.g., DL/S/NO) to improve the narrative flow of the manuscript. Excessive abbreviations can hinder the reader's understanding and disrupt the overall coherence of the text.

Overall, I believe this manuscript constitutes a valuable contribution to the field of animal personality and behavioral syndromes in non-model small mammal species.

Reviewer's Responses to Questions

**Comments to the Author**

1. Is the manuscript technically sound, and do the data support the conclusions?

Reviewer #1: Partly

Reviewer #2: Yes

2. Has the statistical analysis been performed appropriately and rigorously? 

Reviewer #1: Yes

Reviewer #2: Yes

3. Have the authors made all data underlying the findings in their manuscript fully available?

Reviewer #1: Yes

Reviewer #2: Yes

4. Is the manuscript presented in an intelligible fashion and written in standard English?

Reviewer #1: Yes

Reviewer #2: Yes

5. Review Comments to the Author

Reviewer #1: In the Introduction the personality and behavioural sindrome were thoroughly discussed and the cited references were more than enoughto define the experimental problem adreesed by the research.

Although the dataset is quite small (12 and 9 individuals, respectively) the experimental procedure was correct, and described in details allowing thus the repeatibility of the experiment. Each survey was perfomed twice, at a 3-days interval, as suggested by literature. The M&M part is clear and the protocol is well-described. The chosen statistical analyses are sound and well-supported by the references.

The results are clearly shown, and no further analyses are required. In the Discussion the findings are assessed, and supported by the references. The alternative hypothesis suggested for the results contradicting the behavioural sindrome concept is acceptable.

The existence of personality traits in both species of gerbils is an interesting result, but the main concern of the present research is the small number of individuals included in the study, as also recognized by the authors (rows 674-696). The low numerosity could have affected the results, that shown some inconsistencies related to behavioural syndrome. Although the manuscript is overall interesting, the research could have been partly wrecked by the small dataset. I could suggest to include the part dealing with the issue of the small dataset in the M&M section, and then evaluate any potential effect on the present findings in the Discussion.

Reviewer #2: Comments for the Authors - Peer Review of Manuscript PONE-D-23-40950

Having thoroughly reviewed the manuscript titled "Multi-assay approach shows species-associated personality patterns in two socially distinct gerbil species," I find no significant methodological difficulties or other issues that require immediate attention.

The articulation of objectives appears accurate and aligns well with the introduction, methods, and results. The interpretation of the findings not only prompts further investigation but also encourages a more in-depth analysis of various aspects related to animal personality and behavioral syndromes in small mammals, particularly rodents. To substantiate the social niche specialization hypothesis, additional studies are warranted, delving into the mechanisms of phenotypic integration and unraveling the evolutionary paths leading to the development of behavioral syndromes.

While the discussion section is comprehensive, I recommend a careful review to potentially condense its length. Certain parts are challenging to follow, and a more streamlined approach could enhance readability. Additionally, Figure S1 should be incorporated into the main text, as it significantly aids in comprehending the performed behavioral trails.

In the Materials and Methods section, consider minimizing the use of abbreviations (e.g., DL/S/NO) to improve the narrative flow of the manuscript. Excessive abbreviations can hinder the reader's understanding and disrupt the overall coherence of the text.

Overall, I believe this manuscript constitutes a valuable contribution to the field of animal personality and behavioral syndromes in non-model small mammal species.

6. PLOS authors have the option to publish the peer review history of their article (what does this mean?). If published, this will include your full peer review and any attached files.

Reviewer #1: No

Reviewer #2: No

---

## [Author Response · Author response to Decision Letter 0]

31 Jan 2024

Authors’ Responses (in italic) to Editor’s and Reviewers’ Comments

PONE-D-23-40950

Multi-assay approach shows species-associated personality patterns in two socially distinct gerbil species

Academic editor’s comments

After careful consideration, we feel that it has merit but does not fully meet PLOS ONE’s publication criteria as it currently stands. Therefore, we invite you to submit a revised version of the manuscript that addresses the points raised during the review process.

RE: In the revised MS, we carefully followed PLOS ONE’s publication and style requirements (formatting text, tables, figures, and references, code and data sharing, etc.) and recommendations of the reviewers (see below).

RE: We included the statement on the funders’ role (“The funders had no role in study design, data collection and analysis, decision to publish, or preparation of the manuscript”) and adjusted figures to PloS One’s standards with the help of PACE as recommended.

If applicable, we recommend that you deposit your laboratory protocols in protocols.io to enhance the reproducibility of your results. Protocols.io assigns your protocol its own identifier (DOI) so that it can be cited independently in the future. For instructions see: https://journals.plos.org/plosone/s/submission-guidelines#loc-laboratory-protocols. 

RE: The research protocol is described in detail in the MS, as also was noted by Reviewer #1. So, we don’t think that the depositing protocol is applicable.

Journal Requirements:

RE: DONE

RE: DONE. We provided codes as separate Supporting information file (S2_File. docx) along with the data set (S1_File. csv).

3. Thank you for stating the following financial disclosure:"This research was supported by the Russian Science Foundation (22-14-00223 to AVT, https://rscf.ru/project/22-14-00223/)" 

RE: The funders had no role in study design, data collection and analysis, decision to publish, or preparation of the manuscript

RE: We omitted one reference from the list as redundant (Schradin, C., Lindholm, A.K., Johannesen, J., Schoepf, I., Yuen, C., Konig, B., & Pillay, N., (2012). Social flexibility and social evolution in mammals: A case study of the African striped mouse (Rhabdomys pumilio ). Molecular Ecology, 21(3), 541–553.)

Reviewers' comments:

Reviewer #1

The existence of personality traits in both species of gerbils is an interesting result, but the main concern of the present research is the small number of individuals included in the study, as also recognized by the authors (rows 674-696). The low numerosity could have affected the results, that shown some inconsistencies related to behavioural syndrome. Although the manuscript is overall interesting, the research could have been partly wrecked by the small dataset. I could suggest to include the part dealing with the issue of the small dataset in the M&M section, and then evaluate any potential effect on the present findings in the Discussion.

RE: DONE. As recommended, we have moved the passage explaining the “small data set issue” to the M&M section (Lines 138-142) and discuss its possible effects in the Discussion section (Lines 689-707).

Reviewer #2

While the discussion section is comprehensive, I recommend a careful review to potentially condense its length. Certain parts are challenging to follow, and a more streamlined approach could enhance readability. Additionally, Figure S1 should be incorporated into the main text, as it significantly aids in comprehending the performed behavioral trails.

RE: DONE as suggested. We shortened the Discussion and omitted redundant and repetitive phrases and passages to condense the discussion and make it more streamlined. We included Figure S1 in the main text (as Fig 1), as recommended. 

In the Materials and Methods section, consider minimizing the use of abbreviations (e.g., DL/S/NO) to improve the narrative flow of the manuscript. Excessive abbreviations can hinder the reader's understanding and disrupt the overall coherence of the text.

RE: DONE as suggested. We minimized the use of abbreviations throughout the text.

Reviewer's Responses to Questions

Comments to the Author

1. Is the manuscript technically sound, and do the data support the conclusions?

Reviewer #1: Partly

Reviewer #2: Yes

2. Has the statistical analysis been performed appropriately and rigorously?

Reviewer #1: Yes

Reviewer #2: Yes

3. Have the authors made all data underlying the findings in their manuscript fully available?

RE: DONE. We provided the dataset along with codes as separate Supporting information files (S1_File.csv and S2_File.docx, respectively)

Reviewer #1: Yes

Reviewer #2: Yes

4. Is the manuscript presented in an intelligible fashion and written in standard English?

Reviewer #1: Yes

Reviewer #2: Yes

5. Review Comments to the Author

Reviewer #1: 

In the Introduction the personality and behavioural sindrome were thoroughly discussed and the cited references were more than enoughto define the experimental problem adreesed by the research.

Although the dataset is quite small (12 and 9 individuals, respectively) the experimental procedure was correct, and described in details allowing thus the repeatibility of the experiment. Each survey was perfomed twice, at a 3-days interval, as suggested by literature. The M&M part is clear and the protocol is well-described. The chosen statistical analyses are sound and well-supported by the references.

The results are clearly shown, and no further analyses are required. In the Discussion the findings are assessed, and supported by the references. The alternative hypothesis suggested for the results contradicting the behavioural sindrome concept is acceptable.

The existence of personality traits in both species of gerbils is an interesting result, but the main concern of the present research is the small number of individuals included in the study, as also recognized by the authors (rows 674-696). The low numerosity could have affected the results, that shown some inconsistencies related to behavioural syndrome. Although the manuscript is overall interesting, the research could have been partly wrecked by the small dataset. I could suggest to include the part dealing with the issue of the small dataset in the M&M section, and then evaluate any potential effect on the present findings in the Discussion.

RE: DONE. As recommended, we moved the passage explaining the “small data set issue” to the M&M section (Lines 138-142) and discuss its possible effects in the Discussion section (Lines 689-707).

Reviewer #2: 

Having thoroughly reviewed the manuscript titled "Multi-assay approach shows species-associated personality patterns in two socially distinct gerbil species," I find no significant methodological difficulties or other issues that require immediate attention.

The articulation of objectives appears accurate and aligns well with the introduction, methods, and results. The interpretation of the findings not only prompts further investigation but also encourages a more in-depth analysis of various aspects related to animal personality and behavioral syndromes in small mammals, particularly rodents. To substantiate the social niche specialization hypothesis, additional studies are warranted, delving into the mechanisms of phenotypic integration and unraveling the evolutionary paths leading to the development of behavioral syndromes.

While the discussion section is comprehensive, I recommend a careful review to potentially condense its length. Certain parts are challenging to follow, and a more streamlined approach could enhance readability. Additionally, Figure S1 should be incorporated into the main text, as it significantly aids in comprehending the performed behavioral trails.

RE: DONE as recommended (see above). 

In the Materials and Methods section, consider minimizing the use of abbreviations (e.g., DL/S/NO) to improve the narrative flow of the manuscript. Excessive abbreviations can hinder the reader's understanding and disrupt the overall coherence of the text.

RE: DONE as recommended (see above). 

Overall, I believe this manuscript constitutes a valuable contribution to the field of animal personality and behavioral syndromes in non-model small mammal species.

6. PLOS authors have the option to publish the peer review history of their article (what does this mean?). If published, this will include your full peer review and any attached files.

Do you want your identity to be public for this peer review? For information about this choice, including consent withdrawal, please see our Privacy Policy.

Reviewer #1: No

Reviewer #2: No

---

## [Decision Letter · Decision Letter 1]

29 Feb 2024

Multi-assay approach shows species-associated personality patterns in two socially distinct gerbil species

PONE-D-23-40950R1

Dear Dr. Andrey Tchabosky,

We’re pleased to inform you that your manuscript has been judged scientifically suitable for publication and will be formally accepted for publication once it meets all outstanding technical requirements.

Kind regards,

Xiaona Wang, Ph.D

Academic Editor

PLOS ONE

Reviewers' comments:

Reviewer's Responses to Questions

**Comments to the Author**

1. If the authors have adequately addressed your comments raised in a previous round of review and you feel that this manuscript is now acceptable for publication, you may indicate that here to bypass the “Comments to the Author” section, enter your conflict of interest statement in the “Confidential to Editor” section, and submit your "Accept" recommendation.

Reviewer #1: All comments have been addressed

Reviewer #2: All comments have been addressed

2. Is the manuscript technically sound, and do the data support the conclusions?

Reviewer #1: Yes

Reviewer #2: Yes

3. Has the statistical analysis been performed appropriately and rigorously? 

Reviewer #1: Yes

Reviewer #2: Yes

4. Have the authors made all data underlying the findings in their manuscript fully available?

Reviewer #1: Yes

Reviewer #2: Yes

5. Is the manuscript presented in an intelligible fashion and written in standard English?

Reviewer #1: Yes

Reviewer #2: Yes

6. Review Comments to the Author

Reviewer #1: Since all the suggestions from reviewers were met, no further comments are needed. I can suggest the PlosOne Editors to accept the manuscript in the present form.

Reviewer #2: The manuscript has significantly improved. I appreciate the changes made, particularly in the discussion section. It is now more organized and focused on addressing manuscript-specific issues.

7. PLOS authors have the option to publish the peer review history of their article (what does this mean?). If published, this will include your full peer review and any attached files.

Reviewer #1: No

Reviewer #2: No

---

## [Editor Report · Acceptance letter]

1 Apr 2024

PONE-D-23-40950R1 

PLOS ONE

Dear Dr. Tchabovsky, 

I'm pleased to inform you that your manuscript has been deemed suitable for publication in PLOS ONE. Congratulations! Your manuscript is now being handed over to our production team.

Kind regards, 

on behalf of

Associate Professor Xiaona Wang 

Academic Editor

PLOS ONE